# Serological Immune Response Following ChAdOx1 nCoV-19 Vaccine (Covishield^®^) in Patients with Liver Cirrhosis

**DOI:** 10.3390/vaccines10111837

**Published:** 2022-10-30

**Authors:** Amit Goel, Alka Verma, Prachi Tiwari, Harshita Katiyar, Amita Aggarwal, Dheeraj Khetan, Ravi V. Krishna Kishore, Pankaj Kumar, Thakur Prashant Singh, Sabreena Sheikh, Manas Vaishnav, Piyush Pathak

**Affiliations:** 1Department of Gastroenterology, Sanjay Gandhi Postgraduate Institute of Medical Sciences, Lucknow 226014, India; 2Department of Emergency Medicine, Sanjay Gandhi Postgraduate Institute of Medical Sciences, Lucknow 226014, India; 3Department of Clinical Immunology & Rheumatology, Sanjay Gandhi Postgraduate Institute of Medical Sciences, Lucknow 226014, India; 4Department of Transfusion Medicine, Sanjay Gandhi Postgraduate Institute of Medical Sciences, Lucknow 226014, India; 5Department of Gastroenterology and Human Nutrition Unit, All Indian Institute of Medical Sciences, New Delhi 110029, India

**Keywords:** COVID-19, cirrhosis, compensated, decompensated, Covishield

## Abstract

Introduction: Data are limited on antibody response to the ChAdOx1 nCoV-19 vaccine (AZD1222; Covishield^®^) in cirrhosis. We studied the antibody response following two doses of the ChAdOx1 vaccine, given 4–12 weeks apart, in cirrhosis. Methods: Prospectively enrolled, 131 participants (71% males; age 50 (43–58); alcohol-related etiology 14, hepatitis B 33, hepatitis C 46, cryptogenic 21, autoimmune 9, others 8; Child–Turcott–Pugh class A/B/C 52/63/16). According to dose intervals, the participants were grouped as ≤6 weeks (group I), 7–12 weeks (group II), and 13–36 weeks (group III). Blood specimens collected at ≥4 weeks after the second dose were tested for anti-spike antibody titre (ASAb; positive ≥ 0.80 U/mL) and neutralizing antibody (NAb; positive ≥20% neutralization) using Elecsys Anti-SARS-CoV-2 S (Roche) and SARS-CoV-2 NAb ELISA Kit (Invitrogen), respectively. Data are expressed as number (proportion) and median (interquartile range) and compared using non-parametric tests. Results: Overall, 99.2% and 84% patients developed ASAb (titre 5440 (1719–9980 U/mL)) and NAb (92 (49.1–97.6%)), respectively. When comparing between the study groups, the ASAb titres were significantly higher in group II than in group I (2613 (310–7518) versus 6365 (2968–9463), *p* = 0.027) but were comparable between group II and III (6365 (2968–9463) versus 5267 (1739–11,653), *p* = 0.999). Similarly, NAb was higher in group II than in group I (95.5 (57.6–98.0) versus 45.9 (15.4–92.0); *p* < 0.001), but not between the groups II and III (95.5 (57.6–98.0) versus 92.4 (73.8–97.5); *p* = 0.386). Conclusion: Covishield^®^ induces high titres of ASAb and NAb in cirrhosis. A higher titre is achieved if two doses are given at an interval of more than six weeks.

## 1. Introduction

A significant proportion of patients with a severe coronavirus (COVID-19) infection may die [1]. The risk of death is particularly increased in people with co-morbidities such as older age, obesity, diabetes, acute/chronic kidney disease, chronic obstructive pulmonary disease, coronary artery disease, cancers, and chronic liver disease [1,2,3]. Several antiviral drugs have been used to treat COVID-19, but only a few have exhibited an effect against the virus [4,5,6]. In the absence of effective treatment, preventive measures play a pivotal role in the control of the pandemic. Hand hygiene, social distancing, quarantine, face mask use, travel restrictions, lockdown, and other non-pharmacological public health measures seem to have played a prime role in controlling the pandemic [7]. The joint efforts of the global scientific community, coupled with a public–private partnership, resulted in an array of vaccines using different strategies against COVID-19 [8]. These vaccines have been widely used across the world and possibly saved enormous spending as well as a large number of lives [9].

Cirrhosis is a risk factor for severe COVID-19 disease [10] and in-hospital mortality [11]. However, the survival of the patients with cirrhosis is not affected after recovery [12]. The literature suggests that CLD patients are at 6–8 times higher risk for severe disease, and this risk increases with the increasing severity of liver disease [13]. Guidelines established by the reputed international organizations for the study of liver disease [14,15,16] recommend prioritizing COVID-19 vaccination for CLD patients. Natural immunity is compromised in patients with cirrhosis [17]. Responses to non-COVID-19 vaccines such as pneumococcal [18] and hepatitis B [19] are relatively poor in cirrhosis as compared to healthy controls. A recent review has summarized the available data on COVID-19 vaccines in cirrhosis [20]. 

The ChAdOx1 nCoV-19 vaccine (AZD1222) consists of a replication-deficient chimpanzee adenoviral vector ChAdOx1 containing the full-length SARS-CoV-2 structural surface glycoprotein antigen (spike protein; nCoV-19). It is one of the various vaccines which have shown promising protective efficacy in large human trials [21,22]. This vaccine is available under the brand name Covishield^®^ and has been widely used in India. Over 1500 million doses of Covishield^®^ have been administered to this day in India alone. The Covishield^®^ vaccine has also shown promising results in the low-risk population in India [23]. 

The medical fraternity has widely acknowledged the need for data on response to the COVID-19 vaccine in CLD or cirrhosis patients [24]. The data are very limited on the serological response to the ChAdOx1 nCoV-19 vaccine in cirrhosis patients. Therefore, we studied the humoral immune response in cirrhosis patients who were administered two doses of the ChAdOx1 nCoV-19 vaccine (Covishield^®^) as a part of the national vaccination program.

## 2. Materials and Methods

This prospective observational cohort study was conducted between May 2021 and December 2021. Adult (age >18 years) patients with cirrhosis who attended the outpatients’ services of the participating institutes (Sanjay Gandhi Postgraduate Institute, SGPGI; All India Institute of Medical Sciences, AIIMS, New Delhi) were prospectively screened for eligibility criteria and enrolled after obtaining written informed consent. We included cirrhosis patients who were either unvaccinated for COVID-19 or had received two doses of the Covishield^®^ vaccine. Participants were excluded if they had either (i) symptomatic COVID infection, confirmed with the nucleic acid test, at any time before the vaccination, (ii) received two doses of Covishield^®^ vaccine at an interval of more than 36 weeks or (iii) had received another COVID-19 vaccine in addition to Covishield^®^. We also excluded hemodynamically unstable patients and those with acute hepatic or non-hepatic illness. We did not test the participants for the presence of antibodies against the nucleocapsid protein of SARS-CoV-2 to determine the prior subclinical infection.

Cirrhosis was diagnosed with a combination of findings of relevant history and clinical examination, biochemical investigations, radiological evaluations, endoscopic examination for esophageal or gastric varices as evidence of portal hypertension, and AST-Platelet ratio index (APRI). Hepatic decompensation, as recommended for those with hepatitis B virus-related cirrhosis, was defined by the presence of either (i) serum bilirubin more than 2.5 times the upper limit of normal and prolonged prothrombin time (prolonged by >3 s or international normalized ratio >1.5), (ii) occurrence of ascites, or (iii) of hepatic encephalopathy [25].

Relevant data were collected on a predesigned data collection form. Each participant was given two intramuscular doses of Covishield^®^, each of 0.5 mL. The planned interval between the two doses varied from 4 to 12 weeks according to the contemporary recommendations laid by the Government of India. Many participants were delayed in taking their second dose of vaccine, which allowed us to study the effect of vaccine interval on serological response in cirrhosis patients. The study population was categorized into three groups depending on the time interval between the two doses. The participants were grouped as the ≤6 weeks interval (group I), the 7–12 weeks interval (group II), and those with 13–36 weeks of interval (group III).

The vaccine was administered in in-house vaccination facility, established and maintained according to the standard guidelines laid by the government. All the participants were observed on-site for 30 min after each dose. All the participants were on telephone contact for the next 48 h to report any significant adverse event after vaccination.

Specimen collection: For each participant, a 5.0 mL blood specimen was collected after 4 weeks of administering the second dose of vaccine. Serum was separated by centrifugation at 4000× *g* for 10 min at 4 °C within 1 h of blood collection and was stored in multiple aliquots in deep freezers at −80 °C temperature for serological testing at the end of the study.

Serological testing: Stored sera were tested for anti-spike antibody (ASAb) titre and neutralizing antibody (NAb) using Elecsys Anti-SARS-CoV-2 S (Roche Diagnostics GmbH, Sandhofer Strasse 116, D-68305 Mannheim, Germany) and SARS-CoV-2 Neutralizing Antibody ELISA Kit (Invitrogen, Thermo-Fischer, Catalogue no BMS2326), respectively.

Elecsys Anti-SARS-CoV-2 S immunoassay was used for in vitro quantitative assay for antibodies against spike (S) protein receptor-binding domain (RBD). The test’s principle is based on a double-antigen sandwich assay format on an automated system. The assay has a sensitivity of 98.8% and a specificity of 99.98% (as claimed in packet insert of assay kit). The limit of quantitation for the assay is 0.40–250 U/mL. The specimens with antibody titre above 250 U/mL were serially diluted in 20, 50, and 100 folds to obtain the results within the detection range. The titre which remained above the limit of quantitation after 100-fold dilutions was reported as 25,000 U/mL. The antibody concentrations are expressed as U/mL, and a value ≥0.80 U/mL is considered positive for the anti-SARS-CoV-2 anti-spike antibody. The SARS-C0V-2 Neutralizing Antibody ELISA Kit is a competitive ELISA assay. The specimens with calculated neutralization ≥20% are considered positive. Both the assays were performed following the manufacturer’s recommendations.

### Statistical Analysis

Qualitative and quantitative data are expressed as numbers or proportions and median (interquartile range), respectively. The data between the groups were compared using the Mann–Whitney test. The analyses were done using STATA software, version 16 (StatCorp LLC, College Station, TX, USA). Level of significance was kept at <0.05.

## 3. Results

We included 131 participants from two study centers (73 from Lucknow, India (56%); 58 from New Delhi, India (44%)). The clinical and laboratory characteristics of the study participants are summarized in Table 1. Most participants were from Child–Turcotte–Pugh (CTP) class A (52; 39.7%) and B (63; 48.1%) cirrhosis. Nine participants had autoimmune hepatitis-related cirrhosis (seven patients were on immunosuppression: low dose of prednisolone/azathioprine).

Overall, 130 (99.2%) cirrhosis patients developed anti-spike antibodies and their median antibody titre was 5440 (1719–9980) U/mL. A 67-year-old male with cryptogenic decompensated cirrhosis, who had received two doses at an interval of five weeks and whose antibody response was measured nine weeks after the second dose, failed to develop ASAb. The ASAb titres between compensated (5614 (1899–9229) U/mL) versus decompensated (4863 (1674–10,546) U/mL) cirrhosis were comparable (*p* = 0.981). Similarly, ASAb titres were also comparable between CTP class A (5917 (1759–9601) U/mL), B (4216 (1669–10,014) U/mL), and C (5266 (2603–12,282) U/mL) (*p* = 0.575). Neutralizing antibodies were developed in 110 (84%) patients with cirrhosis. Among those who developed neutralizing antibodies, the percentage of neutralization was 92 (49.1–97.6)%. The results of the neutralizing antibodies were also comparable between compensated (92.4%) versus decompensated (91.3%) cirrhosis (*p* = 0.835) and CTP class A (92.2%), B (87.9%), and C (96.6%) (*p* = 0.152). The titres of ASAb were comparable (*p* = 0.539) between the patients with (7031 (2373–11,277) U/mL) or without (4888 (1719–9654)) comorbidities. Similarly, the percentage of NAb was also comparable (*p* = 0.349) between the participants with (77.7 (12.5–96.5)) or without (92 (63.4–97.6)) comorbidities.

In order to analyze the effect of age on immune response, we categorized the participants into two subgroups, i.e., age <45 years and age ≥45 years. The vaccine-induced humoral immune response was comparable between these subgroups (Appendix A); we also found a comparable immune response (Appendix A) between the participants who were categorized, according to etiology of cirrhosis, as viral (hepatitis B or hepatitis C) or non-viral (all other etiologies).

A total of 23 (17.6%), 52 (39.7%), and 56 (42.8%) participants received the second dose after an interval of ≤6 weeks (group I), 7–12 weeks (group II), and 13–36 weeks (group III), respectively. On comparing among the study groups (Table 2), the ASAb titres among the groups I, II, and III were 2613 (310–7518), 6365 (2968–9463), and 5267 (1739–11,653) U/mL, respectively. 

The ASAb titre was significantly higher among those who received the vaccine at an interval of 7–12 weeks compared to those who received it at 6 weeks or less (*p* = 0.027). The ASAb titre was not different between the groups who had an interval of 7–12 weeks or 13–36 weeks (*p* = 0.999) (Figure 1).

Similarly, the percentage of antibody neutralization was significantly higher in the group who was vaccinated at an interval of 7–12 weeks as compared to those who received two doses at an interval of 6 weeks or less (*p* < 0.001), though it was not different between the groups with intervals of 7–12 weeks versus 13–36 weeks (Figure 2). 

The spearman correlation coefficient of association between the intervals between the doses (as a continuous variable) and ASAb titer and Nab percentage were 0.126 and 0.227, respectively.

Barring local pain and low-grade fever, none of the participants developed any significant adverse events during the 48 h of follow-up which required medication or hospitalization.

Of the 131 participants, 4 (3.1%) had a COVID-19 infection after receiving the second dose of the vaccine. All four had a mild infection, did not require hospitalization, and recovered completely. These four participants had received the vaccine doses at an interval of 4, 12, 15, and 18 weeks and their ASAb titres were 11,095, 9654, 13,110, and 13,387 U/mL, respectively.

## 4. Discussion

The present study included cirrhosis patients of various etiologies who received two doses of the ChAdOx1 nCoV-19 vaccine (AZD1222) over 4–12 weeks: 99% and 84% of the participants developed anti-spike antibodies (ASAb) and neutralizing antibodies (NAbs), respectively, with high median titres. The interval between the two doses markedly influenced the ASAb titre and NAb percentage. In contrast, the antibody response was not affected by the etiology or severity of underlying cirrhosis. The antibody response was markedly better in the participants who had received two doses at an interval of more than 6 weeks compared to those given at ≤6 weeks. The serological response was not enhanced further by increasing the second vaccine interval from 7–12 weeks to 13–36 weeks.

Among patients with cirrhosis, bacterial/fungal/superimposed viral infections are associated with increased mortality risk. The available literature suggests poor outcomes in patients with cirrhosis who develop a COVID-19 infection [3,26]. Factors associated with poor outcomes in patients with cirrhosis include advanced age, alcohol-related etiology, increased liver disease severity, and presence of acute or chronic liver failure. Other risk factors for poor outcomes include diabetes, obesity, and chronic kidney disease frequently seen in cirrhosis patients [1].

Large studies suggest that the outcomes in cirrhosis patients can be improved with vaccination by reducing the rate of COVID-19 infection, its severity, need for hospitalization [27], and death [28]. Multiple vaccines are available and have been shown to produce an immune response that effectively reduces the severity of COVID-19 [29]. The immune response to all the COVID-19 vaccines has been shown to be lower in the immunocompromised population, such as organ transplant recipients [30] and those on maintenance hemodialysis [31]. The presence of cirrhosis is also an acquired state of immune deficiency [17], which may prevent the induction of protective immunity following vaccination.

A large multicentric Chinese study, which administered two doses of whole virion Chinese vaccine in a large cohort of patients with CLD, showed the development of NAb in 77% of those with CLD compared to 90% of healthy controls [32]. The response was comparable to the 84% NAb seen in our study. Similar to our results, they also found comparable immunogenicity among people with noncirrhotic CLD, compensated cirrhosis, and decompensated cirrhosis. The data regarding the immune responses following other types of COVID-19 vaccines, such as Pfizer-BioNTech BNT162b2 or the Moderna mRNA-1273 vaccines, in cirrhosis patients are disparate with one showing reduced response [33], and the other showing that the response is not reduced in the presence of cirrhosis [34].

The ChAdOx1 nCoV-19 vaccine is a Chimpanzee adenovirus vector vaccine which contains the SARS-CoV2 spike protein, similar to the AZD1222 COVID-19 vaccine manufactured by AstraZeneca. In India, it is manufactured by the Serum Institute of India, Pune, and marketed under the brand name of Covishield^®^. The Covishield^®^ vaccine is widely used in the Indian population, including those with liver cirrhosis. The safety, immunogenicity, and protective efficacy of the vaccine has been established in non-cirrhotic, immunocompetent populations of different ethnicities [22]. The vaccine was developed with the wild-type strain of COVID-19, but we conducted our study at a time when the Delta variant wave was widely occurring in the country. A change in variant, despite a good serological response, may compromise the clinical effectiveness of the vaccine. The data on the immune response to the ChAdOx1 nCoV-19 vaccine in the immunocompromised population is limited to those with Crohn’s disease [35] or HIV [36], with conflicting results. The immune response seems to be reduced in people with Crohn’s disease [35] but not in HIV-infected patients with undetectable viremia [36].

The data are limited on the immune response following the administration of the COVID-19 vaccine in patients with cirrhosis. Among the various types of COVID-19 vaccines available in the world, the mRNA vaccine (BNT162b2, Pfizer-BioNTech) is one of the most widely used vaccines. Data from other studies suggest that both the humoral [37], as well as T-cell immune responses [38] following the mRNA vaccine (BNT162b2, Comirnaty, Pfizer-BioNTech) are relatively poor in those with cirrhosis as compared to healthy controls.

Our results suggest that the response to vaccination in patients with cirrhosis is not affected by etiology, gender, age, and underlying disease severity. The results are reassuring in that the vaccine is immunogenic in cirrhosis patients. We found that the ASAb titre as well as the NAb percentage were markedly lower in those who had received the two doses at an interval <6 weeks. The second dose of the vaccine primarily act as a booster for immune response. The booster dose is known to enhance the levels of NAb [39]; furthermore, large data pooled from four randomized clinical trials also suggested that more than 12 weeks of interval between the two doses induces better immunity than a shorter interval [40].

Our study had certain strengths. First, this is the first report on ChAdOx1 nCoV-19 vaccine response in a cirrhotic population; second, we included a good number of participants; third, we included participants with a wide spectrum of liver disease severity and etiology; fourth, we included a quantitative assay for neutralizing antibodies. The presence of neutralizing antibodies suggests the protective effect of vaccine-induced anti-spike antibodies. Our study also reported the effect of dosing intervals on immune response in cirrhosis.

Thrombocytopenia is common in cirrhosis patients, particularly those with advanced cirrhosis. Vaccine-induced thrombotic thrombocytopenia has been reported as one of the adverse effects of the ChAdOx1 nCoV-19 vaccine and can have serious outcomes. Most vaccine-induced thrombocytopenia is immune-mediated and manifests itself in the first week following vaccination [41]. In our study, we did not monitor our patients’ platelet counts following vaccination.

The limitations of the study include fewer participants with CTP class C; lack of data on pre-existing ASAb or NAb secondary to subclinical COVID-19 infection before vaccination; the participants were not followed to study the durability of the vaccine-induced immune response and clinical effectiveness.

## 5. Conclusions

In conclusion, a two-dose schedule of the ChAdOx1 nCoV-19 vaccine induces good antibody response in patients with cirrhosis. The antibody response to the vaccine is not affected by the etiology or severity of underlying cirrhosis. The antibody response appears to be better if the doses are administered at an interval of more than six weeks.

## Figures and Tables

**Figure 1 vaccines-10-01837-f001:**
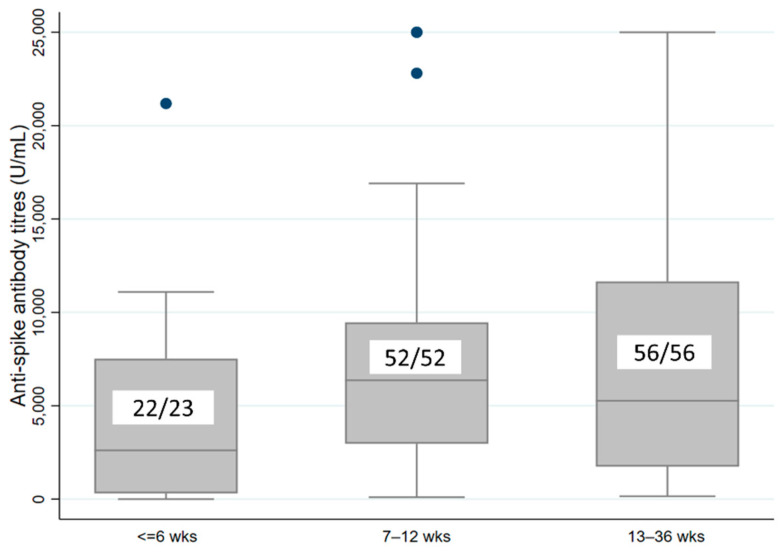
Boxplot comparing the COVID-19 anti-spike antibody titre (U/mL) between the participants who received two doses of vaccine at different intervals. The numbers inside the boxplot show the participants who developed antibodies (numerator) and total number of participant (denominator) in each group.

**Figure 2 vaccines-10-01837-f002:**
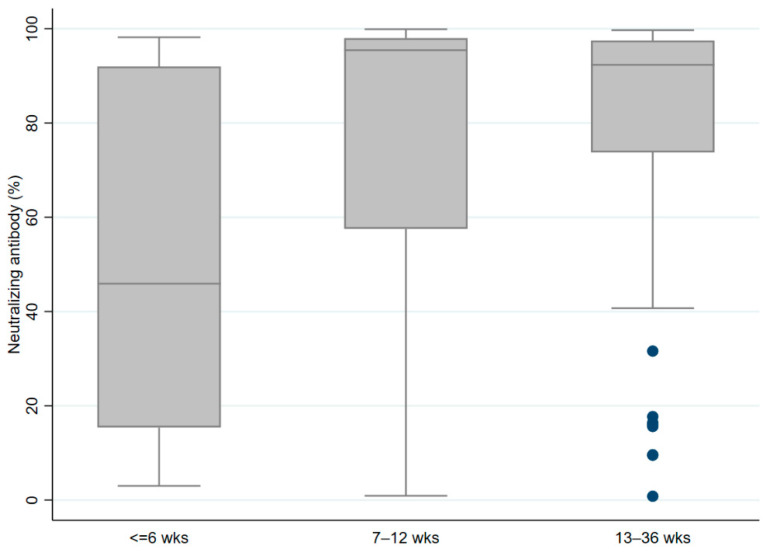
Boxplot comparing the COVID-19 neutralizing antibody percentage between the participants who received two doses of vaccine at different intervals.

**Table 1 vaccines-10-01837-t001:** Characteristics of the study participants (n = 131).

	Values			
Variable	Overall	<6 Weeks Interval between the Doses (Group I), n = 23	7–12 Weeks Interval between the Doses (Group II), n = 52	13–36 Weeks Interval between the Doses (Group III), n = 56
Males (%)	93 (71)	20 (87)	41 (79)	32 (57)
Age (years)	50 (43–58)	58 (44–64)	48 (40–57)	51 (44–56)
Associated conditions				
Diabetes mellitus (DM)	13 (9.9)	3	5	5
Hypertension (HTN)	3 (2.3)	1	1	1
DM + HTN	2 (1.5)	1	0	1
Etiology of cirrhosis				
Alcohol-related etiology	14 (10.7)	0	7	7
Hepatitis B virus	33 (25.2)	7	11	15
Hepatitis C virus	46 (35.1)	5	19	22
Cryptogenic	21 (16.0)	7	7	7
Autoimmune hepatitis	9 (6.9)	2	4	3
Others	8 (6.1	2	4	2
Severity of liver cirrhosis				
Decompensated cirrhosis	80 (61.1)	15 (65)	27 (52)	38 (68)
Child–Turcotte–Pugh score	7 (6–8)	7 (5–8)	7 (6–8)	7 (6–9)
Child–Turcotte–Pugh Class				
CTP A	52 (39.7)	10	24	18
CTP B	63 (48.1)	12	21	30
CTP C	16 (12.2)	1	7	8
Hemoglobin (g/dL)	12.3 (10.7–13.8)	12.9 (10.9–13.6)	12.3 (10.6–13.2)	12.3 (10.5–14.0)
While cell counts (×1000/mm^3^)	4.9 (3.9–6.8)	4.7 (3.9–6.5)	5.0 (4.2–6.9)	5.0 (3.7–6.9)
Platelets counts (×1000/µL)	110 (71–140)	110 (75–150)	110 (80–139)	101 (66–144)
Serum creatinine (mg/dL)	0.9 (0.8–1.1)	0.9 (0.8–1.0)	0.9 (0.7–1.0)	0.9 (0.8–1.1)
Total serum bilirubin (mg/dL)	1.2 (0.8–2.0)	1.0 (0.6–1.7)	1.3 (0.9–2.5)	1.2 (0.8–1.8)
Alanine aminotransferase (IU/L)	35 (29–55)	38 (31–55)	33 (28–56)	37 (30–53)
Aspartate aminotransferase (IU/L)	48 (34–69)	44 (27–66)	44 (35–68)	52 (38–74)
Serum total protein (g/dL)	7.7 (7.2–8.2)	7.8 (6.9–8.2)	7.5 (7.1–8.2)	7.8 (7.3–8.5)
Serum albumin (g/dL)	4.1 (3.5–4.5)	4.3 (3.6–4.7)	4.1 (3.5–4.5)	4.0 (3.6–4.4)
International normalized ratio (INR)	1.28 (1.1–1.56)	1.22 (1.10–1.54)	1.28 (1.10–1.48)	1.3 (1.07–1.59)
Liver stiffness as measured by transient elastography (kPa) #	25.9 (15.9–43.5)	25.1 (15.8–38.5)	27.4 (17.3–44.9)	24.9 (15.6–45.1)
Interval between two doses (weeks)	12 (9–19)	4 (4–5)	12 (11–12)	22 (16–24)
Interval between second dose and specimen collection (weeks)	8 (5–12)	8 (9–12)	7 (4–12)	7 (5–11)

Categorical data are presented as number (proportions); Numerical data are expressed as median (interquartile range); # liver stiffness values were available for a total of 110 participants (22 group I, 44 group II, and 44 group III).

**Table 2 vaccines-10-01837-t002:** Comparison of COVID-19 anti-spike antibody titre and percentage of neutralizing antibody between the groups who were given two doses of vaccine at different intervals.

Study Group	Anti-Spike Antibody Anti-Body Titre	*p* Value		Neutralization Percentage	*p* Value	
Group I: Interval ≤6 weeks	2613 (310–7518)	0.027		45.9 (15.4–92.0)	<0.001	
Group II: Interval 7–12 weeks	6365 (2968–9463)		0.999	95.5 (57.6–98.0)		0.386
Group III: Interval 13–36 weeks	5267 (1739–11,653)			92.4 (73.8–97.5)		

The titre of anti-spike antibody is expressed as U/mL and proportion of participants with neutralizing antibody is expressed as %. Quantitative data are expressed as median (interquartile range).

## Data Availability

The detailed data are available upon reasonable request to the corresponding author.

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
