# Peer review of "Serological Immune Response Following ChAdOx1 nCoV-19 Vaccine (Covishield^®^) in Patients with Liver Cirrhosis"

_vaccines, 2022, doi:10.3390/vaccines10111837_

Round 1
Reviewer 1 Report
There are too many abbreviations without explanation such as on l. 26, 98, 99, 148, or 158. Several sentences ought to be rewritten, such as: l. 240: "vector ChAdOx1, which contains the gene of the full-length structural surface glycoprotein"; l.247: "HIV-infected people whose viral load is controlled"; l. 260-262: "population of reasonable sample size, with inclusion of patients...of both ASAb and NAb.. The NAb response reflects the vaccine protective effectiveness". Fig 1 shows that only 26 of 56 particiants in group III showed anti- spike Ab ! Why is this never said and discussed in the text? Table 1 is full of contradictions in the numbers reported: between columns 1 and 4 (white cell counts, platelet counts, INR...) as well as between columns 1 and 2 ((bilirubin, aspartate aminotransferase, total protein...). How come conversion titers of 77.7% and 92% can be said to be comparable to each other (l. 166)? The list of literature references is at times bizarre: see l. 317 for example! or l. 399. Lancet should be written with a capital L (l. 368).
Author Response
We thank the reviewer for their commnets which helped us to improve the manuscript. We have provided the response in word file.

Reviewer 2 Report
A nice study. A few suggestions.
Can you make it clear to the reader that the vaccine is based on the original COVID strain while the study was conducted during an outbreak of Delta variant.
Table 1 some lines do not seem to line up correctly. Perhaps add lines to guide the eye.
Can you add some discussion regarding the low % neutralization in Group 1.
According to my research (about to be submitted) the age profile for Delta mortality is markedly different to that of the original strain.
No need to cite this paper it is the concept regarding age profile between the vaccine and Delta that is relevant.
Can you add some analysis relating to antibody and neutralization by age.
Can you do some analysis to see if Hep B&C had any effect on the outcome.
Can you do some analysis of antibody and neutralization with time between doses as a continuous variable.
Can you also look at whether the month of initial vaccination had any effect on the results.
As it were, make a good paper even better by delving deeper into the data available.
Perhaps add these as an Appendix.
Author Response

(The authors gave the same response as above.)
